# Therapeutic Drug Monitoring of Ivacaftor, Lumacaftor, Tezacaftor, and Elexacaftor in Cystic Fibrosis: Where Are We Now?

**DOI:** 10.3390/pharmaceutics14081674

**Published:** 2022-08-11

**Authors:** Eva Choong, Alain Sauty, Angela Koutsokera, Sylvain Blanchon, Pascal André, Laurent Decosterd

**Affiliations:** 1Service and Laboratory of Clinical Pharmacology, Department of Laboratory Medicine and Pathology, Lausanne University Hospital and University of Lausanne, 1011 Lausanne, Switzerland; 2Service of Pulmonology, Adult Cystic Fibrosis Unit, Pourtalès Hospital, 2000 Neuchâtel, Switzerland; 3Unit of Adult Cystic Fibrosis Unit and CFTR-Related Disorders, Division of Pulmonology, Lausanne University Hospital and University of Lausanne, 1011 Lausanne, Switzerland; 4Service of Pediatrics, Pediatric Pulmonology and Cystic Fibrosis Unit, Lausanne University Hospital and University of Lausanne, 1011 Lausanne, Switzerland

**Keywords:** TDM, therapeutic drug monitoring, plasma level, PK/PD, dose–effect relationship, cystic fibrosis, LC–MS/MS, CFTR modulators, caftor, ivacaftor, lumacaftor, tezacaftor, elexacaftor

## Abstract

Drugs modulating the cystic fibrosis transmembrane conductance regulator (CFTR) protein, namely ivacaftor, lumacaftor, tezacaftor, and elexacaftor, are currently revolutionizing the management of patients with cystic fibrosis (CF), particularly those with at least one *F508del* variant (up to 85% of patients). These “caftor” drugs are mainly metabolized by cytochromes P450 3A, whose enzymatic activity is influenced by environmental factors, and are sensitive to inhibition and induction. Hence, CFTR modulators are characterized by an important interindividual pharmacokinetic variability and are also prone to drug–drug interactions. However, these CFTR modulators are given at standardized dosages, while they meet all criteria for a formal therapeutic drug monitoring (TDM) program that should be considered in cases of clinical toxicity, less-than-expected clinical response, drug or food interactions, distinct patient subgroups (i.e., pediatrics), and for monitoring short-term adherence. While the information on CFTR drug exposure–clinical response relationships is still limited, we review the current evidence of the potential interest in the TDM of caftor drugs in real-life settings.

## 1. Introduction

Cystic fibrosis (CF) is an autosomal recessive genetic disease caused by variants of the gene encoding the cystic fibrosis transmembrane conductance regulator (CFTR) protein, which affects about 1 of 2700 newborns [1]. The most frequent variant is *F508del*. which is found in 85% of patients [2]. Until recently, treatments of the disease were mostly symptomatic focusing on the consequences of the disease (e.g., mucolytics, antibiotics, pancreatic enzymes, etc.).

Following the discovery of the CFTR gene 30 years ago, great hopes were placed on gene therapy, which is still the subject of significant research, yet without any clinical application being planned in the near future. On the other hand, in the last decade, several molecules called “CFTR modulators” (also nicknamed “caftors”), which partially restore the activity of the CFTR protein, have been developed and are now increasingly used for alleviating the clinical conditions of many CF patients.

These drugs may remedy, in part, the intracellular destruction and/or the malfunction of the CFTR protein and reveal spectacular benefits in terms of respiratory function, nutrition, and quality of life for individuals with CF. The clinical profile is reported to be safe, with most adverse effects being mild to moderate. These new medical breakthroughs are, however, extremely expensive (≈CHF 170.000/year/patient) and target only certain *CFTR* variants [3,4], with limitations regarding the age of the patient and the clinical severity of the disease. Their precise mechanisms of action are yet unknown, and currently, four caftors are registered, all developed by the same pharmaceutical company.

Ivacaftor (VX-770, IVA) is the first caftor and was launched in 2012, marketed as Kalydeco^®^. It is the only registered so-called “potentiator” and targets the *G551D* variant and other variants that affect the gating of CFTR [2]. In order for IVA to have an effect, CFTR proteins must be present on the cell surface. It binds and potentiates CFTR function by promoting decoupling between ATP hydrolysis and gating cycles [4,5,6]. IVA is prescribed as monotherapy for certain CFTR variants or in combination with corrector(s).

To date, three drugs designated as “correctors” are registered and used only in combination with IVA.

Lumacaftor (VX-809, LUM) is a first-generation corrector. It acts similarly to a chaperone, which influences the folding of CFTR in the *F508del*-expressing cell line, resulting in the stabilization of CFTR conformation and translocation to the surface. The combination LUM/IVA (Orkambi^®^) is only approved for *F508del* homozygous patients [3,7]

The second drug, tezacaftor (VX-661, TEZ), in vitro improves the processing and translocation of normal CFTR and certain variants, which leads to an increase in mature CFTR protein on the cell surface. The combination TEZ/IVA (Symkevi^®^, Symdeko^®^) is approved for homozygous and heterozygous *F508del* variants in combination with a specific “residual function” variant in the second *CFTR* allele [3,4,8].

Finally, elexacaftor (VX-445, ELX), a third-generation corrector, has only very recently become available (end of 2019 in the USA and early 2021 in Europe). It is exclusively used in a combination with TEZ and IVA. ELX in vitro improves the processing and transport of CFTR protein variants but binds to other sites of the CFTR protein than TEZ. The new three-drug combination ELX/TEZ/IVA marketed as Kaftrio^®^ or Trikafta^®^ has been reported to provide a more pronounced functional improvement in *F508del* and other variants than that observed with TEZ/IVA [3,4].

TEZ, ELX, and IVA are extensively metabolized by cytochromes P450 3A (CYP3A), characterized by significant variability in expression and activity levels that are notably influenced by environmental factors, and are also likely to be inhibited or induced by various drugs and xenobiotics.

Blood concentration measurement has become one of the very relevant clinical tools to optimize the therapeutic use of critical drugs through adjustment of drug exposure via a therapeutic drug monitoring (TDM) program. The criteria for drugs to be candidates for TDM include significant interindividual pharmacokinetic (PK) variability, poorly predictable from individual patients’ characteristics [9,10], and plasma concentration-response and/or toxicity relationships, defining the plasma concentration ranges associated with optimal efficacy and minimal toxicity.

As the information on CFTR drug exposure–clinical response relationships is still scarce, we aimed to perform a comprehensive review of the current lines of evidence for the potential interest in TDM of caftor drugs by exploiting (i) the data on PK variability and drug exposition retrieved from registration files (Table 1 and Table 2) and (ii) PK data currently reported in a limited number of publications, case series, case reports, and conference abstracts from *real-life* settings (Table 3).

## 2. Methods

For this review, we searched PubMed for publications and conference proceedings. We used the search terms “ivacaftor”, “tezacaftor”, “lumacaftor”, “elexacaftor”, “cystic fibrosis”, and “CFTR modulators” in combination with the specific terms “therapeutic drug monitoring (TDM)”, “area under the curve (AUC)”, “plasma level”, “plasma concentration”, “pharmacokinetic”, “dose–response”, and “dose–response relationship”, covering the literature from 2012 (i.e., first caftor launched) to 31 May 2022.

Besides the registration files, any studies (i.e., observational, case series, and case reports) were selected if participants received at least one marketed caftor, either from a real-life setting or trials published after the latest caftor registration (i.e., 2021), or any studies including children below the minimum recommended age in the registration file (i.e., Trikafta < 6 years). Supportive data from applications to drug registration agencies (FDA, EMA) regarding the PK, pharmacodynamics (PD), and PK/PD studies of CFTR modulators were also included.

## 3. Results

The PK parameters of the four currently approved caftors are given in Table 1 and Table 2.

### 3.1. Pharmacokinetics of CFTR Modulator Drugs

In general, PK parameters are calculated based on the concentrations measured in blood or plasma, compartments that are easily accessible and minimally invasive for patients. Presently, very little information is available on the plasma exposition and steady-state plasma concentrations of caftors in monotherapy and in the different combinations (i.e., ivacaftor, ivacaftor–lumacaftor, ivacaftor−tezacaftor, and ivacaftor−tezacaftor−elexacaftor) achievable in patients under the currently recommended dosage regimens.

In fact, out of a total of 57 studies summarized in a systematic review on the real-world outcomes of IVA, the first and most studied caftor, none have analyzed IVA plasma levels as an outcome [23]. Currently, for the newest marketed combination ELX/TEZ/IVA, there are barely any published real-world observational PK studies [24,25]. Two large ongoing multicenter observational studies, namely RECOVER (Ireland and the UK) and PROMISE (USA), will shed some light on this combination. Blood collection for biomarker analyses is planned in the RECOVER study but, to the best of our knowledge, does not include caftor drug plasma levels [12]. However, the quantification of caftor levels in these collected plasma samples would be feasible, offering the opportunity to perform invaluable retrospective PK analyses for caftors, provided that the time after dose (the interval between last drug caftor intake and blood sampling) has been recorded.

The treatment outcome for caftors depends on several factors such as the severity of the disease, the presence of comorbidities, and also certainly on the circulating plasma concentrations in individuals with CF. The PK parameters for these four caftors are reported in Table 1 and Table 2. An important interindividual PK variability has been reported, for instance, in patients receiving the ELX/TEZ/IVA combination (cf. standard deviation (SD) values in Table 2) [4]. Conversely, the steady state C_max_ for IVA/LUM found in a real-life setting, i.e., outside the stringent frame of clinical trials, was reported to be up to 10 times lower than that of a single-dose level in the labeling information. Moreover, the observed LUM exposure in CF patients was found to correspond to half of that measured in healthy controls (Appendix A) [14,26].

While age and weight [11], for instance, in children, are known to impact the plasma concentrations of caftors, their actual area under the curve (AUC), which constitutes an index of overall plasma exposure, appears to be increased when taken with fat-containing food for all caftors, except TEZ, according to the labeling information [11]. An increase in the AUC of up to four-fold was reported for IVA when taken with fatty meals. Moreover, in their international multicentric clinical PK study, Hanafin et al. noticed notable differences in C_max_ values for IVA and LUM in the various participating centers from different continents, but also among neighboring countries, suggesting that the type of food and socio-cultural eating habits might also modulate caftors’ PK [14].

A number of drug–drug interactions (DDIs) and overlapping drug-related adverse events (AE) have already been documented for caftors. The strong CYP3A inhibitor itraconazole, an antifungal agent, increases by 4- and 15.6-fold the AUC of TEZ and IVA, respectively [27], while some moderate CYP3A inhibitors such as ciprofloxacin showed no apparent alterations in caftor plasma levels. Alternately, coadministration with the strong CYP3A inducer rifampicin significantly reduced the AUC of IVA by ≈89%. Rifampicin is also expected to reduce the exposure of the other CYP3A substrates, namely LUM, ELX, and TEZ. Such lower exposures would result in suboptimal concentrations, and thus, the manufacturer does not recommend this coadministration [4].

Further, because the four current caftors are highly bound (>99%) to plasma proteins, in vitro studies have raised concern for possible drug–drug competition for plasma protein binding sites resulting in an increase in the unbound fractions (i.e., the free pharmacologically active species circulating in plasma), thus leading to the modulation of treatment response. In vitro protein binding competition studies between IVA and albumin and α1-glycoprotein acid in the presence of common comedications, including ibuprofen, loratadine, and montelukast, showed that IVA could strongly be displaced from plasma protein sites [28].

In conclusion, the PK variability of CFTR drugs is recognizably significant, but its impact on treatments’ tolerability and clinical response, the prevalence of toxicity, and the likelihood of, generally unrecognized, drug underexposure in patients remains as yet largely unknown.

### 3.2. Pharmacokinetics–Pharmacodynamics of CFTR Modulator Drugs

The caftor dose–response relationships with the usual CF disease parameters (i.e., body mass index (BMI), forced expiratory volume in one second (FEV1), nasal potential difference (NPD), and sweat chloride concentration) have mainly been studied in dose-escalation regimen carried out in phase II studies with adult CF patients carriers of different genotypes.

A trend of increased response with higher doses was reported for IVA, LUM, and TEZ monotherapy, while no distinct dose–response was observed for ELX over the studied 50–200 mg dosage range.

For IVA, application files to registration agencies have defined the CFTR level at the 90% maximal effect concentrations (EC_90_) with respect to sweat chloride concentration and FEV1. A regimen of IVA 150 mg BID yielded a steady-state plasma C_min_ level of approx. 0.25 μg/mL, corresponding to values ≥EC_90_ for FEV1 and ≥EC_84_ for sweat chloride endpoints, respectively [6].

Conversely, the EC_50_ values of LUM and TEZ were estimated to correspond to plasma C_min_ of 4.5 and 0.5 μg/mL, respectively, using the sweat chloride concentration. A greater reduction in sweat chloride concentration with increasing LUM plasma levels has been reported in the FDA application files. Presently, no in vivo studies have evaluated the EC_50_ of ELX (in vitro EC_50_ is 0.99 μg/mL) [4].

High variability in treatment response has been found in patients with the same *CFTR* genotype and dosage regimen [29,30,31], suggesting that interindividual differences in pharmacokinetics per se are likely incriminated to explain, at least in part, such inconstancy in drug response.

Alternately, no association between the blood levels of IVA/LUM (at average 4 h post-dose and C_min_) and clinical response at 6 months was found in 36 CF patients aged ≥12 years [15].

### 3.3. Safety and Adverse Event (AEs) of CFTR Modulator Drugs

An impressive, systematic review of the real-world safety and relation between the dosage of the first CFTR modulators (IVA, LUM) and AEs thoroughly evaluated nearly 70 studies from 2012 to 2020 [13]. The authors summarized the frequency of discontinuation and adverse events (AEs) related to caftors with detailed patient characteristics and drug dosage regimens. The majority of studies were focused on LUM/IVA (69%), and 6% were on the latest marketed combination TEZ/ELX/IVA. Interestingly, only 16% of the studies were carried out on pediatric patients.

Intriguingly, among these considerable volumes of data and studies in patients presenting drug-related AEs, none measured caftor plasma levels. The lack of robust evidence on target levels, validated quantification methods, guidelines to monitor drug levels and poorly described indications for TDM may partly explain this observation.

Nevertheless, of these 68 articles, 10% assessed a dose reduction in the case of AE. The described AE symptoms in cohorts were related to respiratory intolerance (3/20 *F508del* homozygote CF adult patients with LUM/IVA) [32]; chest tightness (2/29 *F508del* homozygote adult CF patients with LUM/IVA [33] and 1/14 *F508del* homozygote pediatric CF patients with LUM/IVA [34]); and undescribed AE (10/116 *F508del* homozygote CF patients ≥12 years old (yo) with LUM/IVA [35]). Finally, a case report described elevated transaminases with subsequent normalization of symptoms (1 adult CF heterozygous for *F508del* and *R117H-7T* with IVA [36]), and in another one, a breast development was reported as a rare dose-dependent AE of treatment with IVA [16]. Alternatively, a case report concerned a CF adult who discontinued LUM/IVA for respiratory dyspnea AE despite having already a half-dose reduction [37].

In approximately two-thirds of the remaining studies, the AE led to the interruption or discontinuation of caftor treatment. The LUM/IVA AE respiratory-related events could be mitigated in some patients by decreasing the dose. Whether the other described AE of this review are dose- or concentration-dependent and whether a dose adjustment may have maintained the treatment in some of these patients are unknown.

Noticeably, certain AEs have been resolved without any dose change: A case series described respiratory symptoms within 6 weeks of LUM/IVA initiation, which returned nearly to baseline after 2 weeks without any dose change [38]. For patients with ELX/TEZ/IVA presenting testicular pain, the symptoms resolved in 1–12 days after the addition of an OTC analgesic during the continuation of their regular dose [39].

Observational studies with *real-life* CFTR modulator safety data have shown higher rates of discontinuation, as well as AE that were rarely observed or not described in the clinical trials, whereby CFTR modulators are generally well-tolerated, except for IVA/LUM associated with a higher respiratory-related AE [29]. For instance, a higher percentage of previously under-reported AE of mental health deterioration was reported in a *real-life* setting [13,40,41]. Indeed, in 266 CF adults being prescribed ELX/TEZ/IVA, 7.1% of patients reported insomnia, anxiety, mental fogginess, and mood problems. The majority of them underwent a dose reduction. While the sweat chloride (as a surrogate of CFTR function) remained corrected, in 10/13 cases, this AE was improved or resolved by dose reduction (half) and psychological support [42].

Studies in target tissues have reported a destabilization of the corrected F508del CFTR by excessive IVA concentrations [43,44,45], which raised the important question of the possible occurrence of supratherapeutic or toxic concentrations of IVA in some patients [13].

In preclinical studies, an IVA threshold for cataract was defined as ≥10 mg/kg/day in a rat model [4], and cataract also constitutes a potential risk identified with the IVA monotherapy [4,13]. The ELX/TEZ/IVA combination demonstrated an overall improved safety profile, with the most commonly reported side effects being creatine kinase (CK) increase, hepatic enzymes elevation, and rash, which were the most frequent reasons for treatment interruption in clinical studies [4]. There is a lack of information on whether these events are related to drug plasma exposure, or whether the early AE-related drug interruption is related to excessive drug concentrations.

The safety profile of the ELX/TEZ/IVA combination has demonstrated the same potential risks that have been reported for other CFTR modulators: liver function test elevation and increased blood pressure.

Relationships between early adverse drug reactions leading to treatment interruption and plasma levels remain, therefore, to be further studied, especially for the latest-generation combination of CFTR modulators.

### 3.4. Special Population

This is of particular concern for CF patients with altered drug metabolism and/or elimination because of drug malabsorption, hepatic or renal impairments, or for patients taking comedications for other comorbidities (i.e., anti-infective and immunosuppressive drugs) with definite risks of reciprocal DDIs, and should it occur, in case of pregnancy. Additionally, limited clinical information and little hindsight exist for elderly or young patients, especially for children with hepatic impairment. Differences in treatment response or adverse drug reactions owing to physiological and metabolic differences (i.e., differences in distribution volume (Vd), enzyme ontology, etc.) may appear in children.

All the above special situations impact drug disposition, resulting in a definite PK variability with altered drug exposition.

There is also a lack of PK information for pediatric patients below a certain age (from <12 years old (yo) for ELX/TEZ/IVA and <6 yo for LUM/IVA and TEZ/IVA to <4 months for IVA; see Table 2) and also for children with hepatic impairment, as the Child–Pugh scale is not applicable; therefore, no guideline are yet available. Moreover, studies on the drugs’ long-term impact on child development are missing.

Despite the lack of data on in utero drug exposure, a case report described two babies born from mothers receiving CFTR modulators without any evidence of congenital malformations or cataracts [46,47]. Rodent models demonstrated a transfer across placenta and breastmilk of about 40% of the maternal plasma levels [48]. For the postnatal period, barely any information is available, except for the first CFTR modulators. The average LUM and IVA levels in breastmilk were 27.1 μg/l (0.06 μM) and 35.3 μg/L (0.09 μM), respectively, and the average infant plasma levels corresponded to 2.7% and 0.4%, respectively, of the maternal plasma levels [49]. A survey reported no complications in 27 infants exposed to IVA through breastmilk, although the extent of breastfeeding and exposure to caftors were not reported [50]. Besides the safety for the infant, the adequate maintenance of the drug response for pregnant mothers is sparse. Vekaria et al. studied the effect of caftors during pregnancy but without monitoring plasma levels [47]. However, alteration in drug exposition during pregnancy is due to physiological modifications in the volume distribution, clearance, inhibition, or induction of various CYPs and other enzymes. For example, in another therapeutic area, the dose of the antiepileptic drug lamotrigine should, on occasion, be tripled during pregnancy to ensure sufficient exposition [51], and benzodiazepine midazolam doubles its oral clearance during pregnancy [52].

### 3.5. A Case for the Therapeutic Drug Monitoring (TDM) of CFTR Drugs

Thus, current CFTR modulators are generally given at fixed standardized dosages in adults or adapted according to age/weight for children, but such a general “one-size-fits-all” approach does not account for the various factors that also contribute to the large interindividual PK variability reported for these drugs.

As previously stated, information on CFTR drug exposure in a real-life setting still remains limited, with only very few data on the actual target plasma levels (Table 3) [17,18,19,20,21,22].

The best time after dose (TAD) for blood sampling, e.g., C_max_, C_min_ (i.e., before next dose), or AUC determination for obtaining clinically relevant TDM information remains to be formally determined. However, the multiple blood sampling required over an entire dosing interval to calculate AUC is not feasible in a routine clinical setting, even more so in an outpatient ward. Consequently, so far, no guidelines integrate TDM, despite the known PK variability reported for the newer CFTR modulators. Nevertheless, the tailored dose adjustment of these new agents may possibly improve their safety without compromising their efficacy.

TDM may prove to be a clinically useful tool to provide better care to CF patients in a number of instances, which are reviewed in Section 3.6 and Section 3.7.

### 3.6. Drug–Drug Interactions

There are a number of alterations in exposure to caftors due to their cytochromes’ P450-dependent metabolism. Many drugs commonly used in CF patients possibly inhibit CYP3A (e.g., importantly by azole antifungals such as itraconazole) or are metabolized by CYP3A (e.g., antibiotics, steroids, and hormonal contraceptives), and/or are substrates of P-gp (e.g., tacrolimus, ciclosporin, and digoxin) [27,53]. These drug associations may cause DDIs, altering thereby either the exposition of caftors (DDI victim) or the coadministered medication (DDI perpetrator) (see Table 1 and Appendix A).

Notably, given the very high cost of these caftor combinations, and also considering non-responder patients, a PK interaction study has been initiated by Liddy et al. aiming at increasing caftor exposure by adding a CYP3A inhibitor as a “PK booster”, as conducted in the past with ritonavir for HIV protease inhibitors in antiretroviral therapies. It has been indeed demonstrated that IVA AUC was significantly increased in association with ritonavir in healthy volunteers [54].

### 3.7. Adherence to Caftor Treatment

The development of this new life-changing and life-saving oral medication with a simple dosing schedule provided the hope that better adherence would be a natural consequence. A former study with an electronic monitoring device that recorded adherence highlighted the fact that the IVA adherence rate was only 61% of the recommended dosing and decreased over time (*n* = 12 CF ≥ 6 yo) [55]. A mean adherence >80% for all CFTR modulators has more recently been reported [56], with a substantial improvement in the mean adherence of up to 94% (SD 12.4%) at 6 months for the newest regimen ELX/TEZ/IVA [57]. However, for those individuals with suboptimal adherence (including the remaining 6% of patients with poor adherence with ELX/TEZ/IVA), dialogue and reciprocal confidence between healthcare providers and CF patients must be improved. In that context, TDM constitutes an important tool contributing to the promotion of better adherence to these costly therapies, especially considering that the interruption of caftors can lead to withdrawal syndromes with acute deterioration [23,58].

### 3.8. TDM in Other Body Compartments

PK parameters are usually calculated based on concentrations measured in blood, or more generally in plasma, which are the compartments easily accessible and limitedly invasive for patients. However, the airway epithelia levels upon using the caftors were not found to be predicted by serum PK. Moreover, as the drugs act intracellularly, there may be an interest to determine their levels at the expected site of pharmacological actions. To this end, analytical methods have been developed for the quantification of IVA not only in plasma but also in epithelial cells [59,60]. An accumulation of IVA in cells as compared to plasma was observed, confirming a previous in vitro report [60]. In this case, IVA levels in the cellular compartment may be higher than those in circulating plasma, which could lead to a level of CFTR restoration distinct from what would be expected by measuring only plasma levels. Clearly, there is an exciting research area ahead for deciphering caftors’ PK/PD relationships at the tissular/cellular levels.

Other analytical methods to measure caftor concentrations in alternate biological matrices including sputum and rectal organoids have also been described [21,61]. At present, the clinical relevance of, and interest in, TDM by using an alternative matrix instead of plasma for the quantification of caftors remains to be investigated.

### 3.9. Importance of Monitoring the Metabolites

In the TDM practice, enzyme phenotyping, namely the metabolite-to-parent drug ratio (MPR), can be used as a direct measurement of metabolizing enzyme activity (i.e., as carried out, for instance, with midazolam MPR for CYP3A phenotyping) [62]. Unusual MPRs allow for the detection of altered CYP-mediated metabolism activity (e.g., drug–drug interactions, pharmacogenetic variants leading to defective or increased enzyme activity, or liver impairment). Except for LUM, which is marginally metabolized, the metabolites of the caftors can also be simultaneously analyzed with the parent drugs, allowing for the direct monitoring of the MPR in patients: a strong CYP3A inhibitor (e.g., azole antifungals) would lead to a marked decrease in MRP, suggestive for the requirement to adjust caftor dosage [63] (see active metabolite(s) in Table 1).

### 3.10. Practical Implementation of TDM for CFTR Modulator Drugs in the Clinical Setting

In the overview of our development of the TDM of imatinib, the very first oral targeted anticancer agent [64], we have already expressed that a similar TDM approach could apply to a wide range of treatments critical for the control of various life-threatening conditions, such as caftors in CF individuals. The TDM development is generally structured along five generic questions: (1) Is the concerned drug a candidate for TDM? (2) What is the usually observed or target range for the drug’s concentration? (3) What is the therapeutic target for the drug’s concentration? (4) How to adjust the dosage of the drug to drive concentrations close to the target? (5) Does evidence support the usefulness of TDM for this drug?

At present, the information compiled in this review provides strong arguments to support that caftor drugs fulfill most, if not all, criteria for a TDM program (question 1). There is also some information, retrieved notably from clinical studies, on the usually observed concentration ranges for caftors (question 2) (see Table 2). Alternately, there is only limited information on the plasma concentration targets of caftors for optimal therapeutic effect, and to the best of our knowledge, this would need to be formally validated clinically (question 3). For addressing question 4, the pharmacological interpretations of TDM would benefit from incoming computer tools of improved user-friendliness and performance, the development of which is underway [65,66]. For instance, the computer application Tucuxi (http://www.tucuxi.ch, accessed on 10 October 2021) aims at helping practitioners in the interpretation of drug concentration measurements by indicating whether the measured drug levels are “expected” by taking into account drug dosage, patient characteristics, intrinsic population variability, and population PK parameters, and proposes dosage adjustments when indicated. Finally, the definitive answer to question 5 “*Does evidence support the usefulness of TDM for CFTR modulator drugs*?” would be formally obtained through a randomized controlled trial comparing the clinical outcome (i.e., percentages of optimal therapeutic responses and AEs occurrence) of patients being offered a TDM-guided caftor dose adjustment versus patients receiving the currently recommended CFTR drug dosage regimens.

## 4. Discussion

Conceivably, the monitoring of caftors in CF patients’ blood (plasma) shall allow healthcare professionals to have access without delay to information on patients’ plasma exposure and, ultimately, to monitor drug plasma levels in case of drug dosage adjustment. This could be useful in case of clinical complications or toxicities. CFTR drug-dose-related adverse effects (e.g., hepatitis, rash and other skin lesions, gastrointestinal problems for some caftors), or DDI issues could also be addressed. To this end, more data on drug exposure particularly from *real-world* settings are urgently required to evaluate whether plasma levels are within the reference range. TDM has a role in preventing drug toxicity in patients unnecessarily exposed to excessive drug plasma concentrations, by adjusting drug dosage accordingly. Additionally, TDM could help to ascertain that a less-than-expected clinical response may not be due to insufficient exposure to drugs, resulting in impaired gastrointestinal absorption or imperfect treatment adherence.

This review compiled the PK data from *real-life* CF patients, which could also be balanced to those reported within the stringent framework of clinical trials in carefully selected patients who poorly reflect the complex situation of real-life patients. The sparse data on the PK/PD relationship should also be gathered. Finally, the overall patient satisfaction, financial burden of treatments, and pharmacogenetic aspects of these treatments should be explored to even better personalize drug regimens.

## 5. Conclusions

In the growing movement of precision medicine, research efforts must, therefore, be pursued to improve the prescription of CFTR modulators not only with regard to clinical efficacy but also according to tolerability, long-term safety, and potential DDIs, possibly modulated by patients’ pharmacogenetic traits. All the above aspects can be addressed more comprehensively by having access to information on actual caftor exposure in patients’ plasma. In this regard, TDM is at the forefront of this trend to personalize treatment to best meet the needs of CF patients.

## Figures and Tables

**Table 1 pharmaceutics-14-01674-t001:** Relevant pharmacokinetics parameters for the 4 currently approved CFTR modulators [11].

	ELX	TEZ	IVA	LUM
**T_max_ (h)**	6	3	4	4
**AUC fold- increased with fat containing food**	1.9–2.5	1	2.5–4	2
**% bound to plasma protein**	99	99	99	99
**Distribution volume (L)**	53.7	82	293	96
**Enzymes/transporters involved in metabolism**	CYP3A, P–gp	CYP3A, UGT	CYP3A	(CYP3A) ^c^
**Active metabolites**	M23–445 similar potency of ELX.	M1–TEZ similar potency of TEZ.	M1–IVA 1/6 potency of IVA.	M1–LUM 1/6 potency of LUM.
	AUC ratio metabolite/parent: 35–50%	AUC ratio metabolite/parent: 35%		
**Half-life (h)**	27	25	15	26
**Elimination**	97% faeces	72% faeces	88% faeces	51% faeces
**Hepatic function ^a^**	Higher exposure of ELX, TEZ, IVA, LUM is expected in patients with moderate (Child–Pugh Class B, score 7 to 9), and severe hepatic impairment (Child–Pugh Class B, score 10–15).			
**DDI Perpetrator ^b^**	n/a	n/a	Weak CYP3A and P–gp inhibitor	Strong CYP3A inducer; CYP2C9 ^d^, CYP2C19 ^d^, CYP2B6 ^d^ and UGT ^e^ inducers
**Victim DDI with strong ^b^** **CYP3A inhibitor**	AUC 2.8x incr.	AUC 4.5x incr.	AUC 11x incr.	n/a
**Victime DDI with strong CYP3A inducer ^b^**	Co-administration of strong CYP3A inducers (ex: rifampicin) is not recommended			n/a

n/a: not applicable. ^a^ Data available only for adult patients, ^b^ See “Section 3.6 Drug–Drug Interactions” for perpetrator and victim DDIs, ^c^ Not extensively metabolized—the majority of LUM is excreted unchanged, ^d^ King et al. 2022 [12], ^e^ Dagenais et al. 2020 [13].

**Table 2 pharmaceutics-14-01674-t002:** CFTR modulator exposure according to labeling information for the four current CF treatments.

**Mean (±SD) PK Parameters of Ivacaftor (IVA) Monotherapy at Steady State** [6]										
**Age** **Groups** **(Years Old)**		**PK ^a^**								
		**IVA**								
		**Dose**			**C_max_**		**C_min_**		**AUC_0–12 h_**	
					**(μg/mL)**		**(μg/mL)**		**(μg·h/mL)**	
2–5 yo (<14 kg) ^b^		50 mg BID			n/a		n/a		10.5 (4.26)	
2–5 yo (≥14 kg) ^b^		75 mg BID			n/a		n/a		11.3 (3.82) ^d^	
6–11 yo ^b^		150 mg BID			n/a		n/a		20.0 (8.33)	
12–17 yo ^b^		150 mg BID			n/a		n/a		9.24 (3.42) ^d^	
≥18 yo	Single dose ^b^	150 mg QD			0.768 (0.233)		n/a		10.6 (5.26)	
	Trial 809-005 ^c^				1.97 (1.04)		1.06 (0.82)		17.7 (11.7)	
	Trial 005 ^c^				1.433 (0.296)		0.69 (0.238)		12.64 (3.72)	
	Trial 008 ^c^				1.39 (0.522)		0.636 (0.293)		11.6 (4.7)	
	Trial 010 ^c^				1.158 (0.485)		0.523 (0.303)		9.544 (4.603)	
Recommended dose for CF adult		150 mg IVA BID								
Accumulation ratio		2.2–2.9								
Time to reach steady state		3–5 days								
**Mean (±SD) PK Parameters of Tezacaftor (TEZ) and Ivacaftor Combination at Steady State** [8]										
**Age** **Groups** **(Years Old)**		**PK**								
		**TEZ**					**IVA**			
		**C_max_**			**AUC_0–24h_**		**C_max_**		**AUC_0–12 h_**	
		**(μg/mL)**			**(μg·h/mL) ^e^**		**(μg/mL)**		**(μg·h/mL) ^e^**	
6–11 yo (<30 kg)		n/a			58.9 (17.3)		n/a		7.1 (1.95)	
6–11 yo (≥30 kg)		n/a			107 (30.1)		n/a		11.8 (3.89)	
12–17 yo		n/a			97.1 (35.8)		n/a		11.4 (5.5)	
≥18 yo		6.52 (1.83)			82.7 (23.3)		1.28 (0.440)		10.9 (3.89)	
Recommended dose for CF adult		Morning: 100 mg TEZ + 150 mg IVA. Evening: 150 mg IVA (except <30 kg/6–11 yo: TEZ 50 mg QD + IVA 75 mg BID)								
Accumulation ratio		2.3					3			
Time to reach steady state		8 days					3–5 days			
**PK Parameters of Lumacaftor (LUM) and Ivacaftor Combination** [7]										
**Age** **Groups** **(Years Old)**		**PK**								
		**LUM AUC_0–12 h_ (μ** **g·h/mL)**			**IVA AUC_0–12 h_ (μ** **g·h/mL)**					
		** *n* **	**Median**	**Mean**	** *n* **		**Median**		**Mean**	
			**(Range)**	**(SD)**			**(Range)**		**(SD)**	
2–5 yo (<14 kg)		20	175 (131, 339)	180 (45.5)	19		4.64 (2.41, 22.75)		5.92 (4.61)	
2–5 yo (≥14 kg)		42	212 (145, 372)	217 (48.6)	42		5.99 (3.09, 12.51)		5.90 (1.93)	
6–11 yo		165	215 (108, 452)	224 (59.1)	161		5.69 (2.16, 20.04)		6.17 (2.68)	
12–17 yo		98	241 (130, 496)	241 (61.4)	98		3.58 (1.78, 10.26)		3.89 (1.56)	
≥18 yo		264	209 (122, 418)	217 (47.9)	264		3.41 (1.35, 17.31)		3.80 (1.94)	
Recommended dose for CF adult		200 mg LUM + 125 mg IVA BID								
Accumulation ratio		1.9			n/a					
Time to reach steady state		7 days								
**Mean (SD) PK Parameters of Elexacaftor (ELX), Tezacaftor and Ivacaftor** [4]										
**Age** **Groups** **(Years Old)**		**PK**								
		**ELX**			**TEZ**			**IVA**		
		**C_max_**	**C_min_**	**AUC ^f^ τ**	**C_max_**	**C_min_**	**AUC ^f^ τ**	**C_max_**	**C_min_**	**AUC ^f^ τ**
		**(μg/mL)**	**(μg/mL)**	**(μg·h/mL)**	**(μg/mL)**	**(μg/mL)**	**(μg·h/mL)**	**(μg/mL)**	**(μg/mL)**	**(μg·h/mL)**
12 to <18 yo		8.40 (1.75)	4.048 (2.076)	149.0 (38.7)	7.00 (1.65)	2.10 (0.816)	96.0 (23.4)	1.15 (0.288)	0.626 (0.263)	10.60 (3.35)
≥18 yo		8.77 (2.16)	5.488 (2.652)	167.0 (50.5)	6.69 (1.39)	2.05 (0.81)	92.4 (23.8)	1.27 (0.353)	0.75 (0.334)	12.10 (4.17)
Recommended dose for CF adult		Morning: 200 mg ELX + 100 mg TEZ + 150 mg IVA (corresponding of 2 pills). Evening: 150 mg IVA								
Accumulation ratio		2.2			2.07			2.4		
Time to reach steady state		≤7 days			≤8 days			≤3–5 days		

n/a: not applicable, PK: Pharmacokinetics, ss: steady state, yo: years old, QD: once a day, BID: twice a day. ^a^ These data were the original ones retrieved from the registration file in 2011. Since then, the FDA successively approved its use in younger age, currently, the use of this product for infants as young as 4 months old was approved in September 2020, ^b^ These data are retrieved from the prescribing information, initial U.S. Approval: 2012, Revised: May 2017, ^c^ Retrieved from selected multi-dose (5–28 days) in healthy subjects from the FDA registration file, ^d^ Stated as similar to the mean AUC in adult patients administered 150 mg BID, ^e^ AUC_0–24h_ for TEZ and AUC_0–12 h_ for IVA, ^f^ AUC_0–24h_ for ELX and AUC_0–12 h_ for IVA.

**Table 3 pharmaceutics-14-01674-t003:** CFTR modulator multi-dose exposition from real-world setting, or trials published after the latest caftor registration (i.e., 2021), or any studies including children below the minimum age recommended in the registration file.

Drug(s)	*CFTR* Genotype	Location	Study Design	Population	Posology	PK Parameters (C_max_, C_min_, AUC, ss)	*n*	PK Profile	Ref.
LUM/ IVA	n/a	Australia, Europe	Multiple dose, multicenter, open, observational trial reflecting a “real-life” clinical scenario	CF ≥ 12 yo	LUM 200 mg BID and IVA 125 mg BID	Median (IQR) LUM C_max_ 503 (415–1700) ng/mL Median (IQR) IVA C_max_ 59 (24–100) ng/mL	60	Yes	[14]
LUM/ IVA	Homozygous F508del patients	France	Observational follow-up after starting LUM/IVA	CF ≥ 12 yo	LUM 200 mg BID and IVA 125 mg BID	Mean (SD) LUM C_min_ 1675 (75), C_4h_ 1826 (136) ng/mL Mean (SD) IVA C_min_ 72 (17), C_4h_ 151 (42) ng/mL	18	No	[15]
IVA	711 + 1G > T and S1251N mutation	Netherlands	Case report: female CF 7.5 yo patient and CF patient	CF ≥ 6 yo	IVA 150 mg BID	Observed C_4h_ range approx. 1–11 μM (392.5–4317.4 ng/mL), Mean C_4h_ 5.03 μM (1974.2 ng/mL)	16	Yes	[16]
IVA	CFTR gating mutation on at least one allele	USA UK Canada	Ongoing multicenter, phase 3, single-arm, two-part Study in CF children	4 to <6 m	IVA 25 mg BID	Median C_min_ 300 ng/mL, Median AUC 5770 ng∙h/mL	6	No	[17]
				6 to <12 m	IVA 50 mg BID	Median C_min_ 365 ng/mL, Median AUC 7600 ng∙h/mL	16		
				12 to <24 m	IVA 50 mg BID	Median C_min_ 383 ng/mL, Median AUC 8900 ng∙h/mL	19		
				12 to <24 m	IVA 75 mg BID	Median C_min_ 451 ng/mL, Median AUC 9600 ng∙h/mL	2		
TEZ/ IVA	At least one Phe508del CFTR mutation	Australia Europe Israel North America	Multicenter, phase 3, 96-week, open-label study at 170 sites	CF ≥ 12 yo	TEZ 100 mg QD and IVA 150 mg BID	PK exposures to TEZ, IVA, and major metabolites were found similar to those observed in other studies, yet PK profiles or plasma levels are not shown	1044	No	[18]
IVA	CFTR gating mutation on at least one allele	USA UK Canada	Multicenter, phase 3, single-arm, two-part study of IVA in CF children	12 to <24 m	IVA 50 mg BID	Mean (SD) C_min_ 440 (212) ng/mL, Mean (SD) AUC 9050 (3050) ng·h/mL	19	No	[19]
				12 to <24 m	IVA 75 mg BID	Mean (SD) C_min_ 451 (125) ng/mL, Mean (SD) AUC 9600 (1800) ng·h/mL	2		
				2 to 5 yo	IVA 50 mg BID	Mean (SD) C_min_ 577 (317) ng/mL, Mean (SD) AUC 10500 (4260) ng·h/mL	9		
				2 to 5 yo	IVA 75 mg BID	Mean (SD) C_min_ 629 (296) ng/mL, Mean (SD) AUC 11300 (3820) ng·h/mL	26		
IVA	n/a	n/a	Case report: CF patient treated with IVA for ≥3 months old	CF n/a yo	IVA 150 mg BID	Range 400–3000 ng/mL, 5/6 patients had significantly higher levels than those reported from pivotal trial for IVA	6	No	[20]
LUM/ IVA	n/a	Netherlands	CF patients sample for applicability of a developed quantification method	CF (n/a yo)	LUM 800 mg/d and IVA 500 mg/d	Plasma C_2.5 h_ IVA 554 ng/mL, LUM 29300 ng/mL Sputum C_2.5 h_ IVA 64.4 ng/mL, LUM 229 ng/mL	2	No	[21]
					TEZ 100 mg/d and IVA 300 mg/d	Plasma C_2 h_ IVA 924 ng/mL, TEZ 4540 ng/mL			
TEZ/ IVA	n/a	Netherlands	Case report: CF patient with non tuberculous mycobacterium therapy	CF 16 yo	TEZ 100 mg QD and IVA 150 mg BID	AUC TEZ 75400 ng∙h/mL AUC IVA 11100 ng∙h/mL	1	No	[22]

ELX: elexacaftor, IVA: ivacaftor, LUM: Lumacaftor, TEZ: tezacaftor; PK: Pharmacokinetics, ss: steady state, yo: years old, m: months old, IQR: interquartile range, n/a: not applicable.

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
