# Peer review of "Therapeutic Drug Monitoring of Ivacaftor, Lumacaftor, Tezacaftor, and Elexacaftor in Cystic Fibrosis: Where Are We Now?"

_pharmaceutics, 2022, doi:10.3390/pharmaceutics14081674_

Round 1

Reviewer 1 Report

The review deals with an extremely important topic. On the one hand, pharmacokinetics is extremely important for almost all drugs. On the other hand, the therapies discussed are very expensive drugs that are actually administered to patients in a very simple "one fits all" manner. Fortunately, relatively few side effects are observed with these drugs. However, not least the examples given in this review and reported from studies are evidence that one should be more sensitive to this issue. For this reason, the present manuscript is an important contribution to the discussion on CFTR modulators. This is all the more true because the mechanism of action of these drugs is still not entirely clear, nor are long-term side effects foreseeable. 

Unfortunately, the discussion about this is pushed into the background because of the excellent therapeutic effects.

The authors have put a lot of effort into reviewing all the literature on this topic and collecting the important information on it.

However, there are some difficulties with this review. 

Major points:

First, a long section begins with "2. Methods" in that the reader is really only expecting the description tools of how the authors created the manuscript. Instead, besides this important information in the first part starting with "2.1. Pharmacokinetics of CFTR modulator drugs", there are a number of further subsections in which the results of the literature study are already considered. In this respect, this actually belongs in the results/discussion section of the review. Why the authors assigned this to the methods section is not clear. However, it is confusing to read and is simply not correct for the classification for this manuscript. However, the division of the subsections as such could remain as part of a discussion section.

The reader would have liked to see a suggested procedure for how the missing PK data could still be collected in another study program.

The mention of the computer-based program Tucuxi from the University Hospital of Lausanne is too long. It should either be shortened or first experiences should be reported in which this program is actually used for the drugs described here. As it is, it sounds too much like an advertisement.

Minor points:

The manuscript should be checked again for spelling errors, for example page 7 line 332 "between" instead of "betwen".

References: Number 7 begins with "---"

Author Response

We thank the reviewer for their valuable comments. Please find below our point-to-point  responses to address them.

Author: This has been corrected in the revised manuscript. Chapters “Pharmacokinetics of CFTR modulator drugs” and on were originally meant to be the results, i.e. chapter 3.1 and on, instead of 2.1 etc. (numbering error)

Reviewer 2 Report

The review is extremely interesting and covers  information on CFTR drugs exposure and clinical response relationships data that are still limited. Moreover it reviews the current evidence of the potential interest of TDM of caftor drugs in the real-life setting.

The review is well written, clear, and follows a good methodological approach.

Some very small corrections:

1.line 314 and 330 some characters are highlighted

2.line 321 to line 326 are written in italics

3.Unify bibliography and correct errors in references # 7, 22, 31 

Author Response

We thank the reviewer for their valuable comments. Please find below our point-to-point  responses to address them.

Author : the following sentence was added: “Though, quantification of caftors levels in these collected plasma samples would be feasible, offering the opportunity to perform invaluable retrospective PK analyses for caftors, provided that the time after dose (interval between last drug caftor intake and blood sampling) has been recorded”. Of course, this would necessitate formal submission to ethics committee for access to patients’ PK data.

Reviewer 3 Report

The manuscript is well written and clearly organized. The subject of this study is very relevant because of the large and further increasing number of CF patients currently treated with caftors. The authors don't consider the previous findings of Dekkers et al. (JCF 2015) about a functional measurent of CFTR modulators in plasma using intestinal organoids. Plasma levels of caftors seems to be not exhaustive for monitoring the therapeutic effects of these drugs while the effective functional drug levels are detectable by Forskolin Induced Swelling in rectal organoids  carrying the  CFTR genotype described as target of the specific caftor(s). Unfortunately no further studies  on this field have been published more recently about that. I agree with the conclusion of this paper: TDM is required for personalized treatment of CF patients. 

Author Response

We thank the reviewer for their valuable comments. Please find below our point-to-point  responses to address them.

The mention of the computer-based program Tucuxi from the University Hospital of Lausanne is too long. It should either be shortened or first experiences should be reported in which this program is actually used for the drugs described here. As it is, it sounds too much like an advertisement.

Author : We have reduced the manuscript in the mentioned aspect.

Minor points:

The manuscript should be checked again for spelling errors, for example page 7 line 332 "between" instead of "betwen".

References: Number 7 begins with "---"

Author : the manuscript was entirely revised for references et spelling errors